# MT-Ranker: Reference-free machine translation evaluation by inter-system ranking

**Ibraheem Muhammad Moosa, Rui Zhang & Wenpeng Yin**
Department of Computer Science and Engineering
Pennsylvania State University
{ibraheem.moosa,rmz5227,wenpeng}@psu.edu

## Abstract

Traditionally, Machine Translation (MT) Evaluation has been treated as a *regression problem*—producing an absolute translation-quality score. This approach has two limitations: i) the scores lack interpretability, and human annotators struggle with giving consistent scores; ii) most scoring methods are based on (reference, translation) pairs, limiting their applicability in real-world scenarios where references are absent. In practice, we often care about whether a new MT system is better or worse than some competitors. In addition, reference-free MT evaluation is increasingly practical and necessary. Unfortunately, these two practical considerations have yet to be jointly explored. In this work, we formulate the reference-free MT evaluation into a *pairwise ranking problem*. Given the source sentence and a pair of translations, our system predicts which translation is better. In addition to proposing this new formulation, we further show that this new paradigm can demonstrate superior correlation with human judgments by merely using indirect supervision from natural language inference and weak supervision from our synthetic data. In the context of reference-free evaluation, MT-Ranker, *trained without any human annotations*, achieves state-of-the-art results on the WMT Shared Metrics Task benchmarks DARR20, MQM20, and MQM21. On a more challenging benchmark, ACES, which contains fine-grained evaluation criteria such as addition, omission, and mistranslation errors, MT-Ranker marks state-of-the-art against reference-free as well as reference-based baselines.[1]

## 1 Introduction

Automatic MT evaluation is crucial to measure the progress of MT systems. Compared to human evaluation, automatic evaluation is much cheaper and less subjective. Thus, progress in MT has been synonymous with achieving a higher BLEU score (Papineni et al., 2002), the most popular automatic MT evaluation metric. BLEU measures the similarity of a machine translation with a reference translation using n-gram precision. Even though it is still the most widely used metric for MT evaluation, it has a low correlation with human judgment (Freitag et al., 2022). This has prompted researchers to design better automatic evaluation metrics for MT (Bojar et al., 2016; 2017; Ma et al., 2018; 2019; Mathur et al., 2020; Freitag et al., 2021b; 2022).

In recent years, the design of automatic MT evaluation metrics has been dominated by the use of large language models (LLMs) trained on synthetic and human-annotated data (Rei et al., 2020b; 2021; Zhang et al., 2020). These metrics can be broadly categorized into *reference-based* approaches that compare the machine translation with reference translations and *reference-free* approaches that score the machine translation with only the source sentence. The latter approach is arguably more useful in practice since reference translations may be unavailable in real-world scenarios. Interestingly, reference-free evaluation has recently become competitive with reference-based evaluation (Rei et al., 2021).

Irrespective of whether an MT evaluation system uses a reference translation, these systems usually produce a quality score for the machine translation. An alternative approach to machine translation

---

[1]https://github.com/ibraheem-moosa/mt-ranker

is pairwise ranking, where instead of predicting the quality score for a single translation, a pair of translations are compared, and a better-worse judgment is given. This approach was initially proposed by (Ye et al., 2007) and later used by (Duh, 2008; Guzmán et al., 2014; 2015; Mouratidis & Kermanidis, 2019). The pairwise ranking approach is sufficient for the most important use case of automatic evaluation metrics: *comparing machine translation systems*.

However, the pairwise ranking approach remains underexplored. Specifically, it has only been applied in the reference-based evaluation scenario. In this work, we focus on the reference-free scenario. Given the source sentence and a pair of translations, our system will predict which translation is better. This formulation has three benefits compared to the existing MT evaluation approaches. (i) **Simplification of the target task:** The pairwise ranking task is more straightforward than a regression-based task. (ii) **Reference-free evaluation:** In practice, references are often unavailable. Additionally, the reference-based evaluation introduces a reference bias (Freitag et al., 2020). A reference-free evaluation system bypasses these issues since it measures the translation quality directly with the source sentence. (iii) **Less reliance on high quality manual annotations:** Collecting consistent quality scores for machine translations is difficult. For reproducible results, it requires collecting 15 direct assessment annotations per translation (Graham et al., 2015). In contrast, relative ranking annotation from direct assessment with a large enough threshold has been used with as few as one annotation (Ma et al., 2018). Generating synthetic data for the pairwise ranking problem is also much easier. Using synthetic data for pretraining is very effective for training evaluation metric systems (Sellam et al., 2020; Wan et al., 2021). For regression-based approaches, this requires generating a synthetic quality score. In contrast, the pairwise ranking approach only requires a better-worse judgment, which can naturally arise from the synthetic data generation technique.

These benefits motivate us to explore the reference-free pairwise ranking approach to machine translation evaluation. We train our system with supervision only from multilingual natural language inference and some synthetic data. Experiments on DARR20 (Mathur et al., 2020), MQM20 (Freitag et al., 2021a), MQM21 (Freitag et al., 2021b), MQM22 (Freitag et al., 2022), and ACES (Amrhein et al., 2022) demonstrate the state-of-the-art performance of our system without touching any task-specific human supervision. Our contributions can be summarized as follows:

- We are the first to model reference-free MT evaluation as a pairwise ranking problem.
- `MT-Ranker` demonstrates state-of-the-art correlation with human judgments across diverse benchmarks without relying on human-annotated data.
- By eliminating the dependency on human-provided reference translations and comparison data, our system exhibits enhanced practical utility.

## 2 RELATED WORK

**Regression-based MT evaluation.** Since the introduction of BLEU (Papineni et al., 2002), it has been the dominant machine translation evaluation metric. Despite its popularity, BLEU has significant shortcomings, such as low correlation with human judgment (Callison-Burch et al., 2006) and reliance on local lexical features. Researchers have developed evaluation metrics that correlate more with human judgment to address these issues. BERTScore (Zhang et al., 2020) used contextual embedding from BERT (Devlin et al., 2019) to produce a similarity score between the reference and candidate translation, addressing two issues of BLEU: it works even if the candidate translation is a semantically equivalent paraphrase of the reference and considers long-distance dependency. In contrast to BLEU and BERTScore, which are unsupervised, other reference-based evaluation metrics such as RUSE (Shimanaka et al., 2018), BLEURT (Sellam et al., 2020), COMET (Rei et al., 2020a;b; 2021) were trained on human annotations of translation quality scores. BLEURT introduced the use of large-scale synthetic data. The COMET family of models used a dual-encoder architecture that separately embeds source, reference, and candidate translation using transformer encoder models, generating features from combining the embeddings. Trained on human annotations, these metrics achieved higher correlations with human judgments. Interestingly, the reference-free COMET-QE (Rei et al., 2021) model achieved comparable performance with reference-based models. UniTE (Wan et al., 2022) combined reference-free, reference-based, and source-reference-combined evaluation into a single model by training on the three settings simultaneously using multitask learning. PRISM (Thompson & Post, 2020) and BARTScore (Yuan et al., 2021) formulated evaluation as text generation and use the perplexity of the candidate translation as the quality

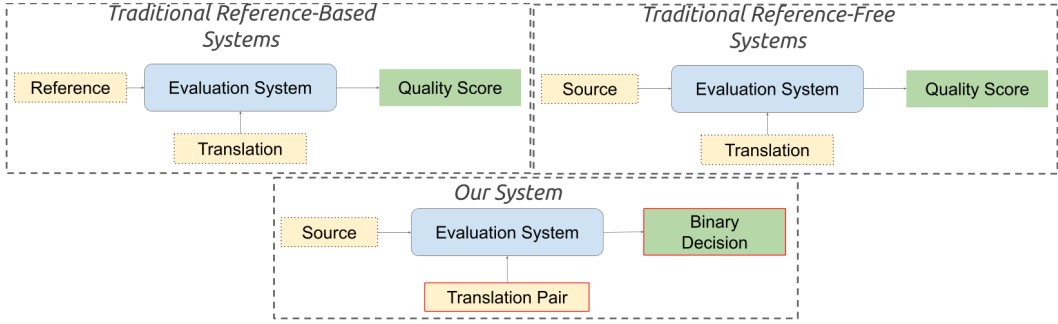

Figure 1: Our system receives a pair of translations and makes a binary decision on which translation has better quality. In contrast, traditional reference-free evaluation systems generate a quality score for a single translation. The main difference between our approach and previous approaches is highlighted in red.

score. These were unsupervised models and performed better than previous supervised models. T5Score (Qin et al., 2022) extended upon BARTScore by discriminative fine-tuning on human annotations. Some recent works (Fernandes et al., 2023; Kocmi & Federmann, 2023; Rei et al., 2023) have explored using GPT-4 (OpenAI, 2023) and PALM-2 (Anil et al., 2023), for translation evaluation, concluding that these models still lag behind on segment-level translation evaluation.

**Pairwise Evaluation.** Reference-based pairwise evaluation of machine translations was first proposed by (Ye et al., 2007). They argued that the ranking annotations have higher inter-annotator agreement than translation quality score annotations. They designed independent features for the reference and each translation. (Duh, 2008) compared ranking with the scoring approach by considering new features constructed from the translation pair under evaluation. (Guzmán et al., 2014) extended upon (Duh, 2008) by considering tree-based features constructed from both the reference and the translation pairs simultaneously. (Guzmán et al., 2015) extended upon (Guzmán et al., 2014) using word embeddings to train neural network models on the pairwise evaluation task. Task-specific pairwise ranking annotations was leveraged for training evaluation metrics (Song & Cohn, 2011; Zhang & van Genabith, 2020). While the models were trained on the comparison data, they ultimately worked in the regression scheme, predicting the quality score for a single translation.

In contrast to all previous approaches, we explore reference-free pairwise evaluation and achieve state-of-the-art correlation with human judgments without using supervised annotation.

## 3 METHODOLOGY

In this section, we first discuss the architecture of `MT-Ranker` and the input formulation (Section 3.1). Then we discuss training with indirect supervision from cross-lingual NLI and weak supervision from synthetically generated data (Section 3.2).

### 3.1 MODEL AND INPUT FORMULATION

We formulate the input to our system as a single text. Given the source sentence $S$ and two translations, $T_0$ and $T_1$, we formulate the input to the model as follows.

$$\textit{Source: S Translation 0: } T_0 \textit{ Translation 1: } T_1 \tag{1}$$

We show an example input to our system in Figure 2. Here, two English translations of a French sentence are being compared. The first translation mistakenly translates the French word *tapis* to the English word *bed*, while the correct translation should be *carpet*. Thus, given this input we would like our system to predict that the second translation is better.

We use the encoder of multilingual T5 (Raffel et al., 2020) as the backbone of our models. The encoder model attends to all the tokens of the source and the two translations simultaneously. We put a mean pooling and a logistic regression layer on top of the encoder model.

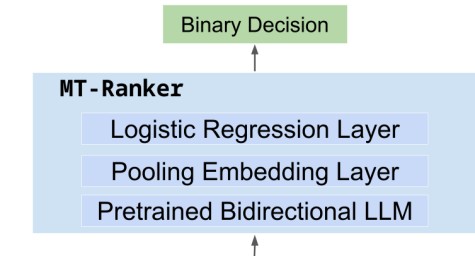

Figure 2: Illustration of the input format and the architecture of `MT-Ranker`. The source sentence and the translation pairs are formatted as a single text input to the system. The bidirectional LLM attends over all the tokens of the source and the translations simultaneously.

## 3.2 MODEL TRAINING

We train `MT-Ranker` on synthetically generated translation pairs where one of the translations can be considered better than the other. Each training sample can be formally represented as follows:

$$(S, (T_0, T_1), y) \tag{2}$$

where $S$ is the source sentence, $(T_0, T_1)$ is the translation pair and

$$y = \begin{cases} 0 & \text{if } T_0 \text{ is better than } T_1 \\ 1 & \text{otherwise} \end{cases} \tag{3}$$

The training is performed in three stages: pretraining with indirect supervision from cross-lingual NLI, fine-tuning on the task of discriminating between human translation and machine translation, and further fine-tuning on weakly supervised synthetic data.

**Stage I: Pretraining with indirect supervision from cross-lingual NLI.** We first pretrain our model on the XNLI (Conneau et al., 2018) dataset. XNLI is a NLI dataset with examples of "Entailed", "Non-entailed", and "Neutral" hypotheses for each premise in fourteen languages. We take the premise in one language and a pair of entailed and non-entailed hypotheses in another language. We consider *the entailed hypothesis to be a better translation of the premise than the non-entailed one*. Although the hypothesis in this dataset does not correspond to actual translations of the premise, this stage serves as an indirect supervision for our model to prefer translations that do not contradict the source sentence. The training samples for this stage can be formally represented as follows:

$$(S, (T_0, T_1), y) \tag{4}$$

$$y = \begin{cases} 0 & \text{if } T_0 \text{ is entailed by } S \\ 1 & \text{otherwise} \end{cases} \tag{5}$$

**Stage II: Discriminating between human translation and machine translation.** At this stage, we construct training pairs based on the assumption that a *human-written reference translation is generally better than machine translations*. The reference translation $R$ is paired with a machine translation, and the task is to predict which one is the reference translation. The training samples for this stage can be formally represented as follows:

$$(S, (T_0, T_1), y) \mid \text{ either } T_0 = R \text{ or } T_1 = R \tag{6}$$

$$y = \begin{cases} 0 & \text{if } T_0 = R \\ 1 & \text{otherwise} \end{cases} \tag{7}$$

We fine-tune our model from Stage I on this task and collect examples of source, reference and machine translations from the Direct Assessment datasets published from 2017 to 2020 (Bojar et al., 2017; Ma et al., 2018; 2019; Mathur et al., 2020). These datasets contain machine translations generated by different machine translation systems for a given source sentence.

**Stage III: Weakly Supervised Training.** One issue with the training pairs used in Stage II is that one of the pairs is the reference translation, which can be of much higher quality than the machine translation. The model never sees translation pairs covering the full spectrum of translation quality. To mitigate this issue, we further fine-tune our model on synthetic data generated by two approaches.

• **Synthetic data from Direct Assessment datasets.** We take samples of machine translation pairs from the Direct Assessment datasets and generate labels for the translation pairs using an unsupervised machine translation evaluation metric $M$. The metric $M$ generates a quality score for each of the translations, which can be used to provide a better-worse judgment for the translation pair. The translation pairs and the target for this stage can be formally represented as follows:

$$(S, (T_0, T_1), y) \tag{8}$$

$$y = \begin{cases} 0 & \text{if } M(S, R, T_0) > M(S, R, T_1) \\ 1 & \text{otherwise} \end{cases} \tag{9}$$

Even though the evaluation metric $M$ has access to the reference $R$, our model does not have access to the $R$ during training or testing. Further, we use an unsupervised metric, BERTSCORE, in this work, so that the model does not have indirect access to human supervision.

• **Synthetic data from machine translation datasets.** Given a translation, examples of worse translations can be generated by perturbing the translation using simple heuristics (Sellam et al., 2020). In general, *the perturbed translation has worse quality than the reference translation.* Thus, the reference translation and the perturbed translation form a training pair for which we can naturally provide a better-worse judgment label. This provides a very simple way to generate a lot of synthetic data for the pairwise evaluation approach. We can formally represent these samples as follows. Given a source sentence $S$, the corresponding translation $T$, and perturbation function $P$, we can generate the following samples

$$(S, (T, P(T)), 0) \tag{10}$$

$$(S, (P(T), T), 1) \tag{11}$$

We generate these types of synthetic data from a subset of the *MT-PRISM* (Thompson & Post, 2020) dataset corresponding to the language pairs represented in the Direct Assessment datasets. *MT-PRISM* is a machine translation dataset collected from diverse sources across 39 languages. We apply the following perturbations to the reference translations to generate examples of worse translations.

- *Word Drop:* We randomly drop 15% of words from the translation. This approach generates examples of worse translations that are not fluent.

- *Word Replacement using Masked Language Model:* We randomly replaced 15% of words in the reference translation using the pretrained XLMRoberta (Conneau et al., 2020) model. The worse translation generated with this approach may be fluent but have a different meaning.

- *Backtranslation:* We translate the sentence to French and back to the original language using the M2M100 (Fan et al., 2021) translation model. The worse translations generated by this approach may preserve the same meaning.

- *Word Replacement after Backtranslation:* We randomly replaced 15% of words from the backtranslated sentence. With this approach, we get examples of worse translations that may not preserve the meaning as well as being worded differently than the original translation.

## 4 EXPERIMENTS

### 4.1 EXPERIMENTAL SETUP

**Datasets.** Our benchmark datasets are the WMT20 Shared Metrics Task dataset (DA20) (Mathur et al., 2020), the MQM20 (Freitag et al., 2021a), the MQM21 (Freitag et al., 2021b), the MQM22 (Freitag et al., 2022) and the ACES (Amrhein et al., 2022) datasets.

• **DA20 (Mathur et al., 2020).** This dataset contains machine translations from 208 MT systems covering 18 language pairs. Each machine translation is annotated with a Direct Assessment quality score by human annotators. Better-worse judgments of translation pairs of the same source are constructed from these DA scores when the scores differ by at least 25 points. The dataset has about 250k such examples of better-worse translation pairs. The large size and the coverage of many language pairs make this dataset highly suitable for evaluating MT evaluation systems.

• **MQM20-22 (Freitag et al., 2021a;b; 2022).** The MQM datasets are constructed by using the Multidimensional Quality Metric (MQM) (Burchardt, 2013) approach by professional translators. These datasets only cover a few language pairs, but the quality score annotationsare of high quality. The MQM20 dataset was constructed by re-annotating a subset of the DA20 dataset. The MQM21 and MQM22 datasets were constructed for the WMT21 and WMT22 Shared Metrics tasks. The MQM20, MQM21, and MQM22 datasets contain 95k, 74k, and 215k examples of better-worse translation pairs, respectively. Even though these datasets cover few language pairs, due to the higher quality of annotations, these datasets are important for benchmarking MT evaluation systems.

• **ACES (Amrhein et al., 2022).** The ACES dataset (Amrhein et al., 2022) is a challenge dataset covering 68 phenomena of translation errors in 10 broad categories: Addition (**A**), Omission (**O**), Mistranslation (**M**), Untranslated (**U**), Do Not Translate (**DNT**), Overtranslation (**Ov**), Undertranslation (**Un**), Real World Knowledge (**RWK**), Wrong Language (**WK**) and Punctuation (**P**). The dataset contains about 36k samples covering 146 language pairs. We use this dataset for fine-grained evaluation of our systems on different challenge scenarios.

**Evaluation Metric.** Since WMT14 (Macháček & Bojar, 2014), the Kendall-like Tau correlation with human judgments has been used to evaluate machine translation evaluation systems. We use this metric to evaluate our system. The formal definition of the metric is given in Appendix A.1

**Baseline Systems.** We compare our system against reference-free evaluation metrics. For evaluation on the WMT20 Shared Metrics Task dataset, our baselines are the two best-performing reference-free systems submitted to WMT20 Metrics Shared Task: COMET-QE (Rei et al., 2020b) and OPENKIWI-XLMR (Kepler et al., 2019) and the reference-free version of T5-SCORE (Qin et al., 2022) which is a recent state-of-the-art system. For the MQM datasets, our baselines are UNITE (Wan et al., 2022), COMET-QE and COMETKIWI (Rei et al., 2022). COMET-QE, and COMETKIWI are the state-of-the-art reference-free evaluation metrics on the MQM21 and MQM22 benchmarks, respectively. We additionally consider KG-BERTScore (Wu et al., 2023) as a baseline for the ACES benchmark since it is the state-of-art on that benchmark.

**Implementation Details.** We consider three variants of the multilingual T5 model with increasing parameter count: Base (290M), Large (600M), and XXL (5.5B). We use the implementation of these models available in the Huggingface library (Wolf et al., 2019). Additional details, including hyperparameters, development set construction for fine-tuning hyperparameters, and training hardware setup, are given in Appendix A.3.

## 4.2 RESULTS

### 4.2.1 RESULTS ON WMT20 SHARED METRICS TASK DATASET

We evaluate our systems on seven X-to-English language pairs and seven English-to-X language pairs from the WMT20 Shared Metrics Task dataset. In addition to the baselines mentioned in section 4, we also show results for SENTBLEU (Post, 2018) and BERTSCORE (Zhang et al., 2020), which are popular reference-based unsupervised machine translation evaluation metrics.

Table 1 shows the segment level Kendall's Tau correlation for all the language pairs. Results for the best-performing model are shown in bold. All our models outperform the baselines. On the X-to-English language pairs, our best-performing model, MT-Ranker-XXL, outperforms the nearest supervised baseline OPENKIWI-XLMR by 4.4 points on average. On the English-to-X language pairs, our best-performing model, MT-Ranker-XXL, outperforms the nearest supervised baseline T5SCORE-XL$_{\text{SUP}}$ by 3.8 points on average.

Table 1: Segment-level Kendall's Tau correlations on language pairs from the WMT20 Shared Metrics Task dataset. Avg. denotes the average score across all language pairs.

| | | cs-en | de-en | ja-en | pl-en | ru-en | ta-en | zh-en | Avg |
|---|---|---|---|---|---|---|---|---|---|
| **X-to-En** | UNSUPERVISED BASELINES | | | | | | | | |
| | SENTBLEU | 6.8 | 41.1 | 18.8 | -2.5 | -0.5 | 16.3 | 9.3 | 12.8 |
| | BERTSCORE | 11.7 | 45.2 | 24.3 | 4.7 | 6.0 | 21.9 | 13.4 | 18.2 |
| | T5SCORE-XL$_{UN}$ | 3.9 | 26.6 | 19.7 | 2.4 | 6.6 | 16.1 | 5.1 | 11.5 |
| | SUPERVISED BASELINES | | | | | | | | |
| | COMET-QE | 9.2 | 40.8 | 15.3 | 4.6 | 10.1 | 16.7 | 9.2 | 15.1 |
| | OPENKIWI-XLMR | 9.3 | 46.3 | 22.0 | 5.9 | 9.2 | 18.8 | 11.5 | 17.6 |
| | T5SCORE-XL$_{SUP}$ | 7.1 | 41.7 | 22.1 | 3.0 | 7.3 | 24.7 | 7.4 | 16.2 |
| | OUR SYSTEMS | | | | | | | | |
| | `MT-Ranker`-Base | 10.7 | 46.8 | 23.2 | 6.8 | **16.9** | 20.9 | 14.8 | 20.0 |
| | `MT-Ranker`-Large | 11.8 | **48.6** | 26.2 | 8.8 | 16.4 | 24.2 | **16.3** | 21.8 |
| | `MT-Ranker`-XXL | **13.1** | 48.5 | **25.6** | **9.2** | 16.3 | **24.9** | 16.2 | **22.0** |
| | | en-cs | en-de | en-ja | en-pl | en-ru | en-ta | en-zh | Avg |
| **EN-to-X** | UNSUPERVISED BASELINES | | | | | | | | |
| | SENTBLEU | 43.2 | 30.2 | 47.9 | 15.3 | 5.1 | 39.5 | 39.7 | 31.6 |
| | BERTSCORE | 51.1 | 39.5 | 53.8 | 28.5 | 20.5 | 60.4 | 41.1 | 42.1 |
| | T5SCORE-XL$_{UN}$ | 19.0 | 20.7 | 47.7 | 11.1 | 12.5 | 39.9 | 18.2 | 24.2 |
| | SUPERVISED BASELINES | | | | | | | | |
| | COMET-QE | 61.3 | 34.6 | 46.7 | 35.8 | 26.4 | 51.2 | 39.8 | 42.3 |
| | OPENKIWI-XLMR | 60.7 | 36.9 | 55.3 | 34.7 | 27.9 | 60.4 | 37.7 | 44.8 |
| | T5SCORE-XL$_{SUP}$ | 62.7 | 40.0 | 58.7 | 37.6 | 28.2 | 66.2 | **45.6** | 48.4 |
| | OUR SYSTEMS | | | | | | | | |
| | `MT-Ranker`-Base | 63.1 | 38.0 | 56.9 | 35.6 | 23.3 | 66.6 | 36.7 | 45.7 |
| | `MT-Ranker`-Large | **69.5** | 46.6 | 60.6 | 42.9 | 27.7 | 69.7 | 41.1 | 51.1 |
| | `MT-Ranker`-XXL | 69.1 | **47.5** | **63.2** | **44.1** | **30.0** | **70.4** | 41.0 | **52.2** |

### 4.2.2 RESULTS ON MQM

We report the Kendall's Tau of baselines and our systems on the MQM benchmark datasets. However, the implementation differs from the one used in (Freitag et al., 2021b; 2022). (Freitag et al., 2021b) flattened the segment-system score matrix before calculating the correlation. Since our system does not predict a quality score for a translation, this approach to Kendall's Tau is unsuitable. Thus, we keep Kendall's tau implementation the same as the one used for evaluation on the DA20 dataset. We note that this implementation is the same as the *segment* averaging approach discussed in (Freitag et al., 2022).

We report the MQM20, MQM21, and MQM22 results in Table 2. We only show results on MQM22 for COMETKIWI since it uses MQM20 and MQM21 as training data. Our best-performing model, `MT-Ranker`-XXL, outperforms the nearest baselines on MQM20, MQM21, and MQM22 by 1.3, 0.4 and 2.8 points respectively.

### 4.2.3 RESULTS ON ACES

In Table 3, we show the performance of our system on the ten categories of the ACES dataset. The final column on the table shows the ACES-Score, a weighted combination of performance in the ten categories. Our system achieves state-of-the-art results in terms of ACES-Score against the previous state-of-the-art system, KG-BERTScore (Wu et al., 2023). Besides the *wrong-language* category of errors that are unsolvable for reference-free systems (Amrhein et al., 2022), our system performs well in all categories. We see especially strong performances by our system in detecting *omission*, *mistranslation*, *do-not-translate*, and *punctuation* errors.

Table 2: Segment-level Kendall's Tau correlation on the MQM datasets. Avg denotes the average correlation achieved by a system across all language pairs on each dataset.

|  | MQM20 | | | MQM21 | | | MQM22 | | | |
|---|---|---|---|---|---|---|---|---|---|---|
|  | en-de | zh-en | Avg | en-de | zh-en | Avg | en-de | zh-en | en-ru | Avg |
| UNITE | 11.1 | 12.2 | 11.6 | 11.4 | 7.3 | 9.3 | 16.3 | 22.4 | 24.8 | 21.2 |
| COMET-QE | 23.4 | 18.6 | 21.0 | **15.2** | 8.8 | 12.0 | 25.3 | 21.6 | 27.2 | 24.7 |
| COMETKIWI | - | - | - | - | - | - | 20.8 | 25.3 | **31.7** | 25.9 |
| MT-Ranker-Base | 19.4 | 13.7 | 16.5 | 13.1 | 7.0 | 10.0 | 15.4 | 18.2 | 18.5 | 17.4 |
| MT-Ranker-Large | 11.6 | 17.9 | 14.8 | 11.0 | **10.6** | 10.8 | 19.8 | 21.9 | 27.2 | 23.0 |
| MT-Ranker-XXL | **25.5** | **19.1** | **22.3** | 14.7 | 10.1 | **12.4** | **24.9** | **26.6** | 31.5 | **27.7** |

Table 3: Kendall's Tau correlation on the ACES dataset.

|  | A | O | M | U | DNT | Ov | Un | RWK | WL | P | ACES-Score |
|---|---|---|---|---|---|---|---|---|---|---|---|
| UNITE | 0.29 | 0.93 | 0.60 | -0.62 | **0.86** | 0.70 | 0.54 | 0.54 | -0.42 | 0.73 | 15.70 |
| COMET-QE | -0.54 | 0.40 | 0.38 | 0.14 | 0.12 | 0.62 | 0.44 | 0.32 | -0.51 | 0.25 | 6.61 |
| COMETKIWI | 0.36 | 0.83 | **0.63** | **0.23** | 0.78 | **0.74** | **0.57** | 0.58 | -0.36 | 0.49 | 16.95 |
| KG-BERTSCORE | **0.79** | 0.81 | 0.49 | -0.46 | 0.76 | 0.65 | 0.53 | 0.49 | **0.31** | 0.26 | 17.49 |
| MT-Ranker-XXL | 0.65 | **0.97** | **0.63** | 0.25 | 0.84 | 0.63 | 0.54 | **0.66** | -0.53 | **0.97** | **18.46** |

# 5 ANALYSIS

We analyze our systems to answer four questions: ($\mathcal{Q}_1$) How much does each stage of training contribute to the final system? ($\mathcal{Q}_2$) How much further can the performance be improved given access to human-annotated training data? ($\mathcal{Q}_3$) Does our system generalize to unseen language pairs? ($\mathcal{Q}_4$) Is there any specific scenario where our systems perform poorly?

**Ablation Study.** To answer $\mathcal{Q}_1$, we study whether the three training stages are necessary for our model's good performance. In Figure 3, we show the performance of our MT-Ranker-Large model on the DA20 dataset after removing different stages from the training process. We plot the average Kendall's Tau across all 14 language pairs. We see that performance drops after removing each of the training stages, indicating the necessity of each training stage. The highest performance drop happens from removing stage 3, showing the effectiveness of our synthetic data generation approach.

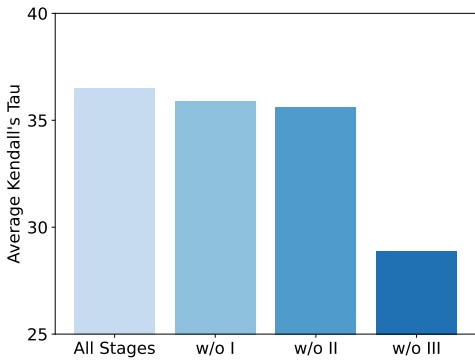

Figure 3: Impact of removing training stages on the performance of MT-Ranker-Large on the DA20 dataset.

**Using Human Supervision.** To answer $\mathcal{Q}_2$, we study whether the performance of our system can be further improved by fine-tuning on human annotations. We construct better-worse translation pairs following the Direct Assessment Relative Ranking approach (Ma et al., 2019) from the Direct Assessment datasets from WMT17, WMT18 and WMT19 Shared Metrics Tasks. We show the results in Figure 4. Across all benchmark datasets, the supervised systems show improved performance over the unsupervised systems.

**Zero-shot performance on unseen language pairs.** Except for the XNLI pretraining stage, our systems are trained on language pairs from the WMT Direct Assessment datasets. To answer $\mathcal{Q}_3$, we investigate whether our system generalizes well to language pairs unseen during training. In Table 4, we show Kendall's Tau correlation for WMT and non-WMT language pairs and their correlation difference on three phenomena from the ACES dataset. The three phenomena and the

Table 4: Correlation difference of our system between WMT and non-WMT language pairs on three phenomena from the ACES dataset.

| | antonym-replacement | | | real-world-knowledge-commonsense | | | nonsense | | | |
|---|---|---|---|---|---|---|---|---|---|---|
| | WMT | Non-WMT | $\delta$ | WMT | Non-WMT | $\delta$ | WMT | Non-WMT | $\delta$ | Avg-$\delta$ |
| MT-Ranker-Base | 0.360 | 0.326 | 0.034 | 0.321 | 0.262 | 0.059 | 0.832 | 0.470 | 0.362 | 0.152 |
| MT-Ranker-Large | 0.728 | 0.621 | 0.107 | 0.490 | 0.490 | 0.000 | 0.804 | 0.531 | 0.273 | 0.127 |
| MT-Ranker-XXL | 0.776 | 0.735 | 0.041 | 0.650 | 0.591 | 0.060 | 0.930 | 0.711 | 0.219 | 0.107 |

Table 5: Kendall's Tau correlation on the untranslated phenomena of the ACES dataset.

| | copy-source | untranslated-vs-ref-word | untranslated-vs-synonym |
|---|---|---|---|
| MT-Ranker-Base | 0.91 | -0.29 | -0.47 |
| MT-Ranker-Large | 0.93 | -0.16 | -0.28 |
| MT-Ranker-XXL | 0.41 | -0.25 | -0.30 |

language pairs were chosen following (Amrhein et al., 2022). We can make the two observations from the results. First, the correlation difference between seen and unseen language pairs is small for the *antonym-replacement* and *real-world-knowledge-commonsense* phenomena. Second, the correlation difference falls with increasing model size, indicating higher generalization as we scale up our models.

**Performance on the Untranslated Phenomena.** To answer $\mathcal{Q}_4$, we focus on the three phenomena of errors in the ACES dataset covered under the category untranslated: *copy-source*, *untranslated-vs-ref-word* and *untranslated-vs-synonym*. Part or the whole machine translation remains untranslated in these types of errors. The untranslated portion can make the translation more similar to the source sentence, and thus, these translation errors can be particularly challenging for reference-free evaluation systems. Table 5 shows the the performance of our systems on these three types of errors. On the *copy-source* errors our systems perform well. On the other two phenomena, our systems perform poorly. This shows our systems' limitations, indicating room for further improvement.

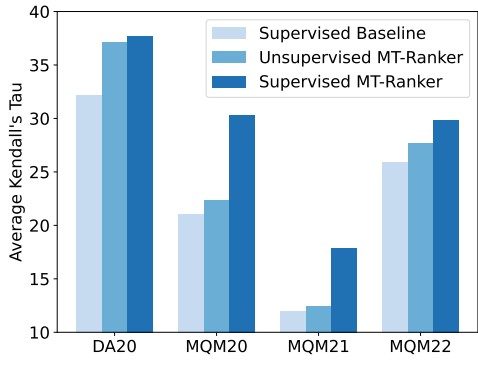

Figure 4: Performance improves on all benchmarks after supervised training.

## 6 CONCLUSION AND DISCUSSION

Machine translation evaluation has been focused on predicting translation quality scores. The alternative approach pairwise evaluation has remained underexplored. The pairwise evaluation approach is sufficient for the most important application of machine translation evaluation systems, comparing machine translation systems. In this paper, we explored pairwise evaluation in the reference-free scenario. This approach can deal with the most practical scenario where reference translation is unavailable. We show that this approach can achieve state-of-the-art correlation with human judgments across five benchmark datasets without requiring any supervision from human-annotated data.

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

# A  APPENDIX

## A.1  FORMAL DEFINITION OF KENDALL'S TAU CORRELATION

The Kendall-like Tau correlation with human judgments can be formally defined as,

$$\tau = \frac{|Concordant| - |Discordant|}{|Concordant| + |Discordant|} \tag{12}$$

where $Concordant$ is the set of all translation pairs where the metric agrees with the human annotators on which translation is better and $Dicordant$ is set of all translation pairs where it disagrees.

## A.2  SYNTHETIC DATASET CREATION

Here we discuss the details of synthetic dataset creation for our third stage of training.

### A.2.1  BERTSCORE

We used BERTScore (Zhang et al., 2020) to generate some of the synthetic data. For the To-English language pairs we use *microsoft/deberta-xlarge-mnli* variant of BERTScore as suggested in the official repository of BERTScore. For the From-English language pairs we use *xlm-roberta-large* version as it achieved the best performance on our development set.

### A.2.2  SUBSAMPLING THE MT-PRISM DATASET

We take at most 25000 samples per language pair from the MT-PRISM dataset. We only consider the language pairs covered in WMT17 to WMT20 Shared Metrics task. In total we have about 4 million samples in this dataset.

### A.2.3  WORD DROP

We used the XLMRoberta (Conneau et al., 2020) tokenizer to tokenize the translation and randomly drop 15% of the tokens from the translation. Here we give an example of sample generated by this approach.

*Source: Doch die Reduktion von CO2 ist eine besonders unwirksame Methode, um den Armen und Hungernden dieser Welt zu helfen.*

*Original Translation: But cutting back on CO2 is a particularly ineffective way to help the world's poor and hungry.*

*Purturbed Translation: But back on CO2 is a particularlyive way to help the world's poor and hungry.*

### A.2.4 WORD REPLACEMENT USING MASKED LANGUAGE MODEL

Similar to Word Drop we use the XLMRoberta (Conneau et al., 2020) tokenizer to tokenize the translation. Then we randomly replace 15% of the tokens using the XLMRoberta model.

Here we give an example of sample generated by this approach.

*Source: Er hasst es, wenn fremde Menschen in seine Welt eintreten.*

*Original Translation: He doesn't seem to like it when other characters intrude on his territory.*

*Purturbed Translation: He doesn't seem to like it when other animals intrude on his territory.*

### A.2.5 BACK TRANSLATION

We take a small subset of 50000 translations from our subsampled MT-PRISM dataset. The translations are then translated to french using the M2M100 (Fan et al., 2021) machine translation model. Then the french translations are back-translated to original language using the same model. We use greedy decoding to generate the translations.

Here we give an example of sample generated by this approach.

*Source: Und trotzdem hat die Ungleichheit zwischen dem ländlichen und dem urbanen Raum zugenommen.*

*Original Translation: And yet, inequality between rural and urban areas has increased.*

*Purturbed Translation: Inequalities between rural and urban areas have increased.*

### A.2.6 WORD REPLACEMENT AFTER BACK TRANSLATION

We again used the XLMRoberta model to perform word replacement. Word replacement was performed on all of 50000 back translated sentences.

Here we give an example of sample generated by this approach.

*Source: Deshalb will die Türkei um jeden Preis die Zeitspanne für mögliche diplomatische Lösungen verlängern.*

*Original Translation: As a result, Turkey wants to extend, at all costs, the time available for diplomacy.*

*Purturbed Translation: As a result, Turkey has to use, at all costs, the time available for diplomacy.*

### A.3 IMPLEMENTATION DETAILS

### A.3.1 DEVELOPMENT SET

To tune hyperparameters we construct a development set from DA17,DA18 and DA19 datasets. We randomly take 50 source sentences per language pair from these datasets. We use the relative ranking samples corresponding to these source sentences as development set. The Kendall's Tau correlation on this development set is used as the validation metric.

### A.3.2 HYPERPARAMETERS

In Table 6 we show the hyperparameters used for training our models. Learning rate and batch size was tuned based on the validation metric. We use early stopping by evaluating the models every 1000 steps of and choose the checkpoint with the highest validation metric.

Table 6: Hyperparameters used for training our models.

| | Batch Size | Learning Rate | #Training Steps (Stage 1) | #Training Steps |
|---|---|---|---|---|
| MT-Ranker-Base | 128 | $5 \cdot 10^{-5}$ | 100k | 20k |
| MT-Ranker-Large | 64 | $5 \cdot 10^{-5}$ | 100k | 20k |
| MT-Ranker-XXL | 32 | $1 \cdot 10^{-5}$ | 20k | 20k |

### A.3.3 TRAINING SETUP

The models were trained A100 GPUs. Each model was trained on a single GPU. We used gradient accumulation for training MT-Ranker-XXL with a batch size of 32. Training MT-Ranker-Base, MT-Ranker-Large, and MT-Ranker-XXL takes about 6, 12, and 52 hours respectively.

### A.4 BASELINE MODELS FOR THE MQM BENCHMARK

Since our evaluation method differs from the standard for the MQM benchmarks we need to reproduce the results of baselines for the MQM datasets. We evaluate against publicly available baselines. Here we give the details of the baseline models.

### A.4.1 UNITE

We chose the Unite-MUP (Wan et al., 2022) model available from the COMET library (Rei et al., 2020a). We use the model in reference-free scenario.

### A.4.2 COMET-QE

We chose the Comet-QE-DA-20 (Rei et al., 2020b) model for evaluation on MQM20 dataset. We chose WMT21-Comet-QE-MQM (Rei et al., 2021) for evaluation on MQM21 and MQM22 dataset. Both of these models are available from the COMET library (Rei et al., 2020a).

### A.4.3 COMETKIWI

We chose the WMT22-CometKiwi-DA (Wan et al., 2022) model available from the COMET library (Rei et al., 2020a).

### A.5 SYSTEM LEVEL EVALUATION

Given the source sentence and a pair of translations our systems predict which translation is better. Thus our systems perform evaluations at the segment level. Here we discuss a straightforward approach to perform system level evaluation from the segment level evaluations.

In system level evaluation we are given the machine translations of a set of source sentences from multiple systems. Since our model predicts a probability of which translation is better we can aggregate these probabilities across all the MT pairs to give a probability score for which MT system is better for each pair of MT systems. Then for each MT system we can average these probability scores to get a score for the system. These scores indicate how likely a MT system is to beat another MT system.

For example, suppose we have three MT systems A, B and C under consideration. We apply our system to each pair and get the scores shown at Table 7. Each cell indicates the probability that the system on the row beats the system on the column. Thus, system A beats system B with a probability of 0.7. We take the average across rows to get the score for each system. Based on the scores we can conclude the ordering $C > A > B$.

We apply this approach to system-level evaluation on the DA20 dataset and achieve competitive results with baseline models. In the Table 8 we report the system-level Pearson correlation of the baseline models and our systems. The Pearson correlation is calculated against the system level gold human evaluation scores.

Table 7: System level evaluation for a hypothetical scenario.

|   | A | B | C | Average |
|---|---|---|---|---|
| A |   | 0.7 | 0.3 | 0.5 |
| B | 0.3 |   | 0.4 | 0.35 |
| C | 0.7 | 0.6 |   | 0.65 |

Table 8: System-level Pearson correlations on language pairs from the WMT20 Shared Metrics Task dataset. Avg. denotes the average score across all language pairs.

|  |  | cs-en | de-en | ja-en | pl-en | ru-en | ta-en | zh-en | Avg |
|---|---|---|---|---|---|---|---|---|---|
| X-to-En | SUPERVISED BASELINES |  |  |  |  |  |  |  |  |
|  | COMET-QE | 75.5 | 93.9 | 89.2 | 44.7 | 88.3 | 79.5 | 84.7 | 79.4 |
|  | OPENKIWI-XLMR | 76.0 | 99.5 | 93.1 | 44.2 | 85.9 | 79.2 | 90.5 | 81.2 |
|  | T5SCORE-XL$_{SUP}$ | 73.0 | 99.5 | 95.9 | 51.3 | 94.0 | 92.0 | 87.9 | 84.8 |
|  | OUR SYSTEMS |  |  |  |  |  |  |  |  |
|  | MT-Ranker-Base | 80.0 | 100.0 | 94.0 | 52.0 | 91.0 | 92.0 | 88.0 | 85.3 |
|  | MT-Ranker-Large | 80.0 | 99.0 | 94.0 | 60.0 | 92.0 | 93.0 | 89.0 | **86.7** |
|  | MT-Ranker-XXL | 80.0 | 99.0 | 94.0 | 58.0 | 90.0 | 92.0 | 88.0 | 85.9 |
|  |  | en-cs | en-de | en-ja | en-pl | en-ru | en-ta | en-zh | Avg |
| EN-to-X | SUPERVISED BASELINES |  |  |  |  |  |  |  |  |
|  | COMET-QE | 98.9 | 90.3 | 95.3 | 96.9 | 80.7 | 88.7 | 37.5 | 84.0 |
|  | OPENKIWI-XLMR | 97.2 | 96.8 | 99.2 | 95.7 | 87.5 | 91.0 | -1.0 | 80.9 |
|  | T5SCORE-XL$_{SUP}$ | 98.5 | 96.3 | 96.4 | 96.8 | 88.7 | 95.2 | 96.0 | **95.4** |
|  | OUR SYSTEMS |  |  |  |  |  |  |  |  |
|  | MT-Ranker-Base | 99.0 | 94.0 | 98.0 | 97.0 | 63.0 | 93.0 | 47.0 | 84.4 |
|  | MT-Ranker-Large | 99.0 | 95.0 | 99.0 | 98.0 | 76.0 | 94.0 | 55.0 | 88.0 |
|  | MT-Ranker-XXL | 99.0 | 95.0 | 99.0 | 98.0 | 75.0 | 94.0 | 57.0 | 88.1 |

## A.6 INCONSISTENCIES IN PAIRWISE RANKING DECISIONS

We have discussed above how to get system level scores with our systems by averaging across the row of pairwise comparison probabilities. With the system-level scores, there are no contradictions. However, there is a chance of contradiction among the better-worse judgments of systems if we do not provide this system level score and rely on each pairwise comparison probabilities to make decisions. To that end, we have checked if there is any inconsistency among the pairwise system ranking generated by our system.

Table 9: Percentage of triples from DA20 where `MT-Ranker`-Large makes inconsistent predictions.

| zh-en | en-de | ru-en | ta-en | en-zh | pl-en | en-pl |
|-------|-------|-------|-------|-------|-------|-------|
| 0.54% | 0.14% | 0.91% | 0.27% | 0.91% | 0.41% | 0.27% |

Among the 14 language pairs on the DA20 datasets we have only found inconsistency on 7 of them. On the following table we show the percentage of triples that show inconsistency. We note the inconsistency level is less than one percent for all the 7 language pairs.

## A.7 RESULTS FOR FURTHER LANGUAGE PAIRS ON WMT20 SHARED METRICS TASK

On our main table for the WMT20 Shared Metrics Task we did not report results for the language pairs *km-en, ps-en, iu-en, en-iu* for brevity of presentation. In Table 10 we report results for these language pairs.

Table 10: Segment-level Kendall's Tau correlations on further language pairs from the WMT20 Shared Metrics Task dataset.

|  | km-en | ps-en | iu-en | en-iu |
|---|-------|-------|-------|-------|
| SUPERVISED BASELINES |  |  |  |  |
| COMET-QE | 14.9 | 9.2 | 3.2 | -6.3 |
| OPENKIWI-XLMR | 24.4 | 10.6 | 5.6 | 6.0 |
| T5SCORE-XL$_{\text{SUP}}$ | 22.5 | 10.6 | 2.6 | -2.6 |
| OUR SYSTEMS |  |  |  |  |
| `MT-Ranker`-Base | 30.3 | 19.0 | 1.6 | **20.3** |
| `MT-Ranker`-Large | **33.4** | **20.1** | 5.2 | 14.7 |
| `MT-Ranker`-XXL | 33.1 | 19.7 | **6.6** | 18.0 |

Our systems strongly beat the baseline systems on these languages even though our systems were not trained on human annotations. For the iu-en and en-iu language pairs we noticed that we filtered out these language pairs also from our training. The strong performance on these two pairs also show the generalization of our system to unseen language pairs during training.

