# OpenReview forum: "MT-Ranker: Reference-free machine translation evaluation by inter-system ranking"
_ICLR.cc/2024/Conference — ICLR 2024 spotlight_

### Official Review · Reviewer_2G4a · 2023-10-26

**Soundness:** 2 fair
**Presentation:** 3 good
**Contribution:** 2 fair
**Rating:** 6
**Confidence:** 3

**Summary:**

This work proposes Comparator, a reference-free MT evaluator that treats the evaluation as a inter-system comparison problem. The model consists of a pre-trained encoder that accepts the concatenation of the source sequence and a pair of translations to compare and a comparator head that pools the output embeddings and produces a binary decision on which translation is better. The model is trained in three stages: XNLI pre-training (preferring entailment over non-entailment), human/machine translation discriminating (preferring human translation over machine translations) and weakly-supervised tuning (pairs of translations judged by BertScore and synthetic data by perturbation). The proposed model is evaluated on various MT eval datasets and the results show its effectiveness and benefits even over supervised baselines.

**Strengths:**

- The proposed method is straight-forward and shows good performance over a range of datasets.
- Some of the indirect and weakly supervised training method is interesting and might be inspiring for future study of MT evaluation.

**Weaknesses:**

- I’m wondering if the comparison is fare for other systems since the proposed model is trained with multiple external resources such as XNLI and especially some data with parallel sentences, and is based on large base pre-trained encoders. I think it would be more convincing if there can be ways to directly compare different evaluation systems (ref-free vs ref-included, score-based vs comparison-based) with the same training resources and base models. (But surely, it would be indeed a benefit if the proposed system can better utilize extra resources.)
- There should be more analysis on the proposed methods (such as those in Section 5). For example, more ablation studies on the training stages and especially the usage of different resources, and more detailed analysis on the metric and perturbation methods in Stage 3. Some of the result and setting sections may be shorten or moved to the appendix.
# --
- (Updates): Most of these concerns are addressed by the authors' responses, and the authors should provide those details in later versions.

**Questions:**

- What pre-trained model did the baseline systems use? Did they use up to XXL models? (Same or similar base models should be utilized for fair comparisons.)
- Are there any ways to convert the relative comparisons to absolute scores? Sometimes, we might still need the scores (for example as rewards for RL).
- In Stage 2 and the first part of Stage 3, references are required for the training purpose. If those two parts are ablated, how would it influence the results?

---

> ### Author Response · Authors · 2023-11-21
> **Response to Reviewer 2G4a**
>
> We thank the reviewer for their valuable feedback. The reviewer appreciated the simplicity and superior performance of our approach. The reviewer also acknowledged the interestingness of our weakly supervised training method.
>
> **Response to the weakness mentioned by the reviewer:**
>
> **I’m wondering if the comparison is fare for other systems…:**
> 1. **Concern about using the external data such as XNLI and MT-Prism:**
> One of our baseline T5Score uses the same machine translation dataset MT-Prism to train their model. Other than SentBLEU all our baselines are based on pretrained-encoders. *It is fair to compare our systems against these systems since our system does not require costly human annotations of machine translation quality scores from humans.* Regarding comparing reference-included and reference-free systems, it is better to have a reference-free systems since in practice references are not always available. The fact that our system does not require the references during inference is a strength of our system.
> 2. **Concern about using the same amount of data as the baselines:**
> Regarding comparing score-based vs comparison-based systems on same training resources, we have argued in our paper that it is much easier to gather synthetic data for comparison based approach, which means comparison based approach has the advantage of having larger synthetic training data. Since it is the advantage of our approach, constraining our approach to have the same amount of training data would be unfair to our systems.
> 3. **Concern about using different pretrained model from baselines:**
> The reviewer also mentioned having the same base model as our baselines. Our best performing baseline T5-Score uses the mT5 model which is the same model we have used. Our second-best performing baseline used the XLM-Roberta-Large model to train their systems. To make sure our performance is not just due to using mT5 model, we have now trained a XLM-Roberta-Large based model using our method. The performance for this model which we call Comparator-XLMR on the DA20 benchmark is shown below. Comparator-XLMR achieves superior performance compared to baselines and similar performance to our Comparator-Large and Comparator-XXL models.
>
> | Model           | cs-en | de-en | ja-en | pl-en | ru-en | ta-en | zh-en | Avg  |
> |-----------------|-------|-------|-------|-------|-------|-------|-------|------|
> | Comparator-XLMR | 9.6   | 47.3  | 26.9  | 10.1  | 16.2  | 23.6  | 14.6  | 21.2 |
>
> | Model           | en-cs | en-de | en-ja | en-pl | en-ru | en-ta | en-zh | Avg  |
> |-----------------|-------|-------|-------|-------|-------|-------|-------|------|
> | Comparator-XLMR | 69.3  | 41.9  | 58.9  | 44.4  | 31.0  | 69.5  | 51.2  | 52.3 |
>
>
>
> **There should be more analysis on the proposed methods…:**
>
> We have performed further ablation studies of our different training stages. We trained ablated versions of our Comparator-Large model by removing different training stages. We show the performance of our ablated models on the DA20 dataset in the following table. Here sub-stage 3.1 refers to using the BERTScore based weak supervision and sub-stage 3.2 refers to the perturbation based weak supervision in our stage 3. The results show the specific contribution of metric and perturbation methods in Stage 3. The BERTScore based weak supervision has more contribution to the final performance. However, all the stages contribute positively to our final performance.
> | All Stages | w/o 1 | w/o 2 | w/o 3 | w/o 3.1 | w/o 3.2 | w/o 1,2 | w/o 2,3 |
> |------------|-------|-------|-------|---------|---------|---------|---------|
> | 36.5       | 35.9  | 35.6  | 28.9  | 28.5    | 35.9    | 35.5    | 9.9     |

---

> > ### Author Response · Authors · 2023-11-21
> > **Continuation of Response to Reviewer 2G4a**
> >
> > **Response to reviewer’s questions:**
> > 1. One of our baseline T5-Score uses an XL sized model. Our other baselines systems use XLM-Roberta-Large and Info-XLM-Large models which have similar size as the mT5-Large model. Our Comparator-Large model which is based on mT5-Large beats all these baselines on the DA20 benchmark. We have also reported results with XLM-Roberta-Large previously in this response which also beats all the baselines. We also want to note that, in contrast to the baselines our system does not need costly human annotation data and is trained on weakly supervised synthetic data.
> > 2. Given a set of machine translations, we can perform an all-pairwise comparison with our system and rank the translations based on how many times a translation is deemed better by our system. This ranking can be used to give a score. However, for training RL systems giving an absolute reward score is not always necessary. Recent NLP models that are trained using RL use human preference judgment annotations where human annotators only give better-worse judgments.[1] Our model can be used as a proxy to human annotators to synthetically generate these better-worse judgments, which then can be used to train the RL models.
> > 3. The references are required in stage 2 and 3 to generate the synthetic data. If we ablate the references completely from our system then we are left with only stage 1. The ablation results shown in the previous table shows that without stage 2 and 3 the performance of the system is very poor. Thus access to the reference translations is required for training of our system. *Our focus in this work is to remove the reliance on costly human annotations of translation quality scores for training machine translation evaluation models.* Collecting these annotations are much costlier than collecting reference translations which can even be performed automatically by bi-text mining. Thus we believe reliance on reference translations during training is not a drawback of our system.
> >
> >
> > [1] Ouyang, Long et al. “Training language models to follow instructions with human feedback.” ArXiv abs/2203.02155 (2022): n. pag.

---

### Official Review · Reviewer_GQxu · 2023-10-31

**Soundness:** 3 good
**Presentation:** 3 good
**Contribution:** 3 good
**Rating:** 8
**Confidence:** 4

**Summary:**

This paper proposes a comparative MT evaluation metric: instead of comparing machine translations to references, it compares multiple machine translations. The model is built on a bidirectional LLM, that encodes pairs of translations, and a pooling and logistic regression layer on top. It is trained with data from crosslingual NLI, pairs of human and machine translations, and synthetically rated or corrupted pairs of translations. Evaluation is done on a set on the WMT20 metrics task, several MQM datasets and ACES, a challenge dataset. The proposed model is compared to previous state-of-the-art reference-free metrics and largely outperforms them across languages, as well as supervised baselines.

**Strengths:**

- Idea and method are simple and well explained. Given that it requires much less costly data than the competing methods, it poses an attractive solution for MT evaluation.
- The reported results are strong, given that they do not require references or direct supervision. The proposed model outperforms both supervised and unsupervised baselines.
- The ablations give an insight into the importance of the different stages and generalization to unseen languages, which allows one to get a more thorough understanding of the method and its benefits.

**Weaknesses:**

The novelty seems limited / overemphasized. In Quality Estimation (QE) reference-free ranking approaches have been used for MT quality estimates before it was re-invented in the context of MT metrics competitions. For example, in the very first QE task in 2012 (https://www.statmt.org/wmt12/quality-estimation-task.html) ranking based evaluations (without references) were already designed, and as a result, ranking based methods have been developed as well (e.g. Avramidis, Eleftherios. "Sentence-level ranking with quality estimation." Machine translation 27.3-4 (2013): 239-256.; Eleftherios Avramidis. 2012. Comparative Quality Estimation: Automatic Sentence-Level Ranking of Multiple Machine Translation Outputs. In Proceedings of COLING 2012, pages 115–132, Mumbai, India.)

**Questions:**

- Can you explain what the sentence “relative ranking annotation from direct assessment with a large enough threshold has been used with as few as one annotation” (Intro) means? Where does the threshold come into play and what does it mean to have one annotation only (I assume one per input, but not only one input)?
- Figure 1 is not adding much, its content is clear from the text. The space could be used to elaborate the connections to QE (see above) and report more empirical results on newer datasets.
- Is there any train/test overlap of the training data of mT5 and the benchmarks’ test data?
- What if references were removed from the fine-tuning sets in Stages 2 and 3? This would be a useful ablation to make the models truly free from references.
- Which of the evaluation differences are significant?
- What if Stages 1&2 are dropped and only Stage 3 is performed? This would further illustrate the importance of that stage and make it clear for future use where most time investment should go.

---

> ### Author Response · Authors · 2023-11-21
> **Response to reviewer GQxu**
>
> We thank the reviewer for their valuable feedback. The reviewer appreciated the simplicity and superior performance of our approach as well as the fact that our system requires much less costly data. The reviewer also appreciated the analyses we performed on our system.
>
> **Response to the weakness mentioned by the reviewer:**
>
> We thank the reviewer for bringing to our attention the related works that we missed. We note the following differences between our approach and the previous literature.
> 1. The previous systems rely on feature engineering and on external parser to generate some of the features. In contrast our approach does not rely on feature engineering and can work directly on the source sentence and the translation pair.
> 2. Our models show generalization to unseen language pairs during inference. In contrast, the previous systems only work on a single language pair.
> 3. Our system does not rely on costly human-annotated quality scores of machine translations and achieves superior performance with weakly supervised data. The previous systems mentioned by the reviewer were trained on human annotations of quality judgments.
>
> **Response to reviewer’s questions:**
> 1. Direct assessment quality scores of machine translations collected from human evaluation is very noisy. It requires at least 15 annotations per machine translation to get reproducible scores. Since it is costly to get so many annotations per translation especially for low resource languages, in practice much less annotations may be available, sometimes as few as one. However, in this scenario suppose translation A gets one annotation of direct Assessment score 80 and translation B gets one annotation of Direct Assessment score 30, the difference in score being 50, we can still conclude that translation A is better than translation B. In previous literature a threshold of 25 score difference has been used to make similar conclusions.
> 2. We think it is important to have the figure since our approach is different from the usual MT evaluation. It is important to convey the difference clearly to readers.
> 3. mT5 was pretrained on the mC4 corpus. The latest data in mC4 corpus is from Common Crawl dump 2020-34 which includes web scrapes upto August 15th 2020. The WMT20 dataset was released on August 23rd 2020. Other benchmark datasets we used were released much later. Thus we are confident that the mT5 models have not seen the benchmark test data.
> 4. The references are required in stage 2 and 3 to generate the synthetic data. If we ablate the references completely from our system then we are left with only stage 1. The ablation results in our paper and in the table later in this response shows that without stage 2 and 3 the performance of the system is very poor. Thus access to the reference translations is required for training of our system. *Our focus in this work is two-fold, i) to remove the requirement of having references during inference and ii) to remove the reliance on costly human annotations of translation quality scores for training machine translation evaluation models.* Collecting these annotations are much costlier than collecting reference translations which can even be performed automatically by bi-text mining. Thus we believe reliance on reference translations during training is not a drawback of our system.
> 5. We performed the PERM-BOTH test following previous literature.[1] The results show that our Comparator-Large and Comparator-XXL model significantly outperforms all baselines on the DA20 benchmark dataset with a p-value < 0.05. On the MQM20 and MQM22 benchmark dataset our Comparator-XXL model significantly outperforms all baselines on en-de and zh-en language pairs with p-value <0.5. Even though our system does not significantly outperform other baselines in other settings, our system does not use any supervised data during training  in contrast to the baselines.
> 6. We have performed further ablation experiments. We show the results in the following table. The results of our ablation show the performance of the system drops by one point to 35.5 when we remove stage 1 and 2 and just perform stage 3.
>
> | All Stages | w/o 1 | w/o 2 | w/o 3 | w/o 3.1 | w/o 3.2 | w/o 1,2 | w/o 2,3 |
> |------------|-------|-------|-------|---------|---------|---------|---------|
> | 36.5       | 35.9  | 35.6  | 28.9  | 28.5    | 35.9    | 35.5    | 9.9     |
>
>
> [1] [A Statistical Analysis of Summarization Evaluation Metrics Using Resampling Methods](https://aclanthology.org/2021.tacl-1.67) (Deutsch et al., TACL 2021)

---

> > ### Comment · Reviewer_GQxu · 2023-11-22
> > **Response**
> >
> > Thank you for the clarifications of my questions and the additional ablation, very helpful.
> > I recommend highlighting the distinction of references during training vs inference as well, and discussing the listed nuances of the novelty over previous methods (and at the same time acknowledging these previous methods more).

---

### Official Review · Reviewer_Qwxh · 2023-11-05

**Soundness:** 4 excellent
**Presentation:** 4 excellent
**Contribution:** 4 excellent
**Rating:** 8
**Confidence:** 4

**Summary:**

In this paper, the authors propose a method to learn pairwise reference-less evaluation of MT. This scenario corresponds to a common, real use where reference translations are mostly unavailable and the interest is mostly in comparing systems, rather than absolute scores. The pairwise ranking is a good framework since it is easier to collect synthetic data with pairwise judgments, as opposed to assigning quality scores to synthetic examples. Pairwise rankings also achieve higher inter-annotator agreement than human judgments. The method proposes a 3-stage pipeline for finetuning with various kinds of synthetic data.

The results show that the approach can achieve state-of-the-art performance comparable or better than supervised approaches. The analysis also shows that adding supervised data can further improve performance modestly. Interpreted in another way, it also means that the synthetic data generated is sufficient to capture most of the attributes of supervised data for pairwise ranking of systems.

**Strengths:**

* The paper is well-written and explains the motivation for the work well.
* The experiments are extensive and establish that reference-less evaluation is very competitive with reference-based metrics.
* The use of synthetic data for pairwise ranking is a very clean way of using synthetic data and helps train high-quality reference-less metrics.

**Weaknesses:**

While pairwise evaluations of systems are useful, a more practical utility would be to rank multiple models. Score-based systems enable that easily. With pairwise ranking-based systems, multiple comparisons have to be run. Every time a new system has to be ranked, it has to be compared with multiple existing systems.

**Questions:**

* Equation 11 should be y=0?
* 3 languages from the WMT-20 Metrics tasks have not been included in the evaluation. It would be good to include those results as well for getting a complete view of the WMT-20 Metrics task.

---

> ### Author Response · Authors · 2023-11-21
> **Response to reviewer Qwxh**
>
> We thank the reviewer for their valuable feedback. The reviewer appreciated that our paper is well written and explains our motivation well. The reviewer also appreciated the extensive experiment results reported in our paper and the use of synthetic data to achieve good performance.
>
> **Response to weaknesses mentioned by the reviewer:**
> 1. In practice we do not have that many systems to compare. Usually it is less than a dozen. In this scenario, our system does not add much overhead. Given the simplicity of our system, not relying on human annotations during training, not requiring access to references and generalization to unseen languages, it is an attractive alternative to existing score-based approach.
> 2. We can provide our models with more than two translations. On the pooling layer we can construct the pooled embedding for all pairs simultaneously. This can be done in parallel with full matrix operations. These pooled embeddings can be sent to classifier head in parallel to get rankings for all the translations. We are currently pursuing this approach to create a more efficient system.
>
>
> **Response to reviewer’s questions:**
>
> 1. The reviewer is correct. We will fix the issue in our revised paper.
> 2. We report the results for the three languages of DA20 in the following table.
>
> | Model            | km-en | ps-en | iu-en | en-iu |
> | ---------------- | ----- | ----- | ----- | ----- |
> | Openkiwi-XLMR    | 24.4  | 10.6  | 5.6   | 6.0   |
> | Comet            | 14.9  | 9.2   | 3.2   | \-6.3 |
> | T5Score-XL       | 22.5  | 10.6  | 2.6   | \-2.6 |
> | Comparator-Base  | 30.3  | 19.0  | 1.6   | 20.3  |
> | Comparator-Large | 33.4  | 20.1  | 5.2   | 14.7  |
> | Comparator-XXL   | 33.1  | 19.7  | 6.6   | 18.0  |
>
> Our systems strongly beat the baseline systems on these languages even though our systems were not trained on human annotations. We unintentionally filtered out these languages in our evaluation. Additionally, for the iu-en and en-iu language pairs we noticed that we filtered out these language pairs also from our training. The strong performance on these two pairs also show the generalizability of our system to unseen language pairs during training.

---

### Official Review · Reviewer_DxQx · 2023-11-05

**Soundness:** 2 fair
**Presentation:** 3 good
**Contribution:** 2 fair
**Rating:** 5
**Confidence:** 4

**Summary:**

This paper proposes a novel reference-free machine translation evaluation method which directly compares the two hypotheses from two systems by pre-trained language models, e.g., mT5.

**Strengths:**

The manuscript is commendably clear in its presentation, providing a lucid explanation of the method's underpinnings and its design rationale. The method itself is logically structured and appears to be grounded in a sound understanding of the underlying technical principles.

**Weaknesses:**

(Main) 1. **Experimental Settings**: Upon meticulous examination, I observe that the experimental setup deviates from the conventional practices of evaluating numerous systems simultaneously. The study opts to assess a custom set of 'Better-worse judgments' between pairs of system outputs. This focus narrows the scope of the evaluation and raises concerns about the validity of the correlation results. The paper's method shows a strong correlation with these judgments but fails to conclusively demonstrate the superiority of one system over another. The more critical challenge lies in integrating these pairwise comparisons into a comprehensive evaluation framework that can handle multiple systems. The current approach's limitations in addressing this challenge may undermine its utility and applicability.

2. **Scope and Generalization**: The narrow focus of the study may limit its applicability beyond its stated domain. The paper could benefit from an expanded discussion on how the proposed method might adapt or extend to other evaluation contexts, such as the assessment of large language models (LLMs). While ICLR might be receptive to specialized domain contributions, the paper's current emphasis suggests that it might find a more fitting audience at a dedicated NLP conference.

**Suggestions**
1. **Contextualizing with Recent Advances**: Recent developments in MT evaluation metrics, such as xCOMET (https://arxiv.org/abs/2310.10482) and SLIDE (https://arxiv.org/pdf/2309.08832.pdf), emphasize the importance of error span evaluations. While it is not mandatory to compare against the latest publications, incorporating insights from these advancements could significantly enhance the robustness and relevance of the proposed method.

2. **Related Work**: I recommend that the authors consider the insights from "DABERTScore" (https://aclanthology.org/2021.acl-short.5.pdf). There are conceptual overlaps between this work and the manuscript under review, which merit a thorough comparison and discussion within the paper to enrich the context and underscore the novel contributions of the proposed method.

**Questions:**

1. How should the paper address the ranking of two MT systems? Is there a more robust method than simply tallying the number of better translations?
2. When scaling up the evaluation to multiple systems, how might one resolve apparent ranking contradictions, such as the scenario where System A outperforms System B, System B outperforms System C, but System C outperforms System A?

These questions are pivotal in addressing the practical implications of the proposed method and its capacity to function in a more complex, real-world evaluation environment. A more thorough exploration of these aspects could substantially strengthen the paper’s contribution to the field.

---

> ### Author Response · Authors · 2023-11-21
> **Response to reviewer DxQx**
>
> We thank the reviewer for their valuable feedback. The reviewer acknowledged the soundness of our approach and the clear presentation of our paper.
>
> **Response to the weakness mentioned by the reviewer:**
>
> 1. A straightforward approach to applying our system to perform system-level evaluation is to compare all machine translation pairs of pairs of MT systems. Since our model predicts a probability of which translation is better we can aggregate these probabilities across all the MT pairs to give a probability score for which MT system is better for each pair of. Then for each MT system we can average these probability scores to get a score for the system. These scores indicate how likely a system is to beat another system. For example, suppose we have three systems A, B and C under consideration. We apply our system to each pair and get the following scores. Each cell indicates the probability that the system on the row beats the system on the column. Thus, system A beats system B with a probability of 0.7. We take the average across rows to get the score for each system. Based on the scores we can conclude the ordering C>A>B.
>
> |   | A   | B   | C   | Avg  |
> | - | --- | --- | --- | ---- |
> | A |     | 0.7 | 0.3 | 0.5  |
> | B | 0.3 |     | 0.4 | 0.35 |
> | C | 0.7 | 0.6 |     | 0.65 |
>
> We apply this approach to system-level evaluation on the DA20 dataset and achieve competitive results with baseline models. In the following table we show the results. The reported values are Pearson correlation with the system level gold human evaluation scores.
>
>
> | Model            | en-cs | en-de | en-ja | en-pl | en-ru | en-ta | en-zh  | Avg EN-X |
> | ---------------- | ----- | ----- | ----- | ----- | ----- | ----- | ------ | -------- |
> | OpenKiwi-XLMR    | 0.972 | 0.968 | 0.992 | 0.957 | 0.875 | 0.91  | \-0.01 | 0.809    |
> | COMET-QE         | 0.989 | 0.903 | 0.953 | 0.969 | 0.807 | 0.887 | 0.375  | 0.840    |
> | T5-Score-XL-SUP         | 0.985 | 0.963 | 0.964 | 0.968 | 0.887 | 0.952 | 0.960  | 0.954    |
> | Comparator-Base  | 0.99  | 0.94  | 0.98  | 0.97  | 0.63  | 0.93  | 0.47   | 0.844    |
> | Comparator-Large | 0.99  | 0.95  | 0.99  | 0.98  | 0.76  | 0.94  | 0.55   | 0.880    |
> | Comparator-XXL   | 0.99  | 0.95  | 0.99  | 0.98  | 0.75  | 0.94  | 0.57   | 0.881    |
>
> | Model            | cs-en | de-en | ja-en | pl-en | ru-en | ta-en | zh-en | Avg X-EN | Avg All     |
> | ---------------- | ----- | ----- | ----- | ----- | ----- | ----- | ----- | -------- | ----------- |
> | OpenKiwi-XLMR    | 0.76  | 0.995 | 0.931 | 0.442 | 0.859 | 0.792 | 0.905 | 0.915    | 0.862       |
> | COMET-QE         | 0.755 | 0.939 | 0.892 | 0.447 | 0.883 | 0.795 | 0.847 | 0.915    | 0.878       |
> | T5-Score-XL-SUP         | 0.73 | 0.995 | 0.959 | 0.513 | 0.94 | 0.92 | 0.879  | 0.915 | 0.935   |
> | Comparator-Base  | 0.8   | 1     | 0.94  | 0.52  | 0.91  | 0.92  | 0.88  | 0.853    | 0.849
> | Comparator-Large | 0.8   | 0.99  | 0.94  | 0.6   | 0.92  | 0.93  | 0.89  | 0.867    | 0.874       |
> | Comparator-XXL   | 0.8   | 0.99  | 0.94  | 0.58  | 0.9   | 0.92  | 0.88  | 0.859    | 0.870       |
>
> 2. Machine translation evaluation is an important field in natural language processing. We have proposed a new framework for MT evaluation that beats state-of-art supervised baselines without using any human-annotated comparison data. Given the simplicity of our approach and the large reduction in human annotation cost, we believe our work is a significant contribution that should be of interest to the wider machine learning community.  We also note that in previous ICLR conferences papers on Dialogue system evaluation has been publishe[1]That being said, we are currently exploring using our approach to other NLG evaluation tasks as future work.
>
> 3. We note that the xCOMET paper was released after we submitted our paper to ICLR and the SLIDE paper was released only 12 days before our submission. We agree that fine-grained error span evaluations are an important direction for MT evaluation systems. We plan to explore this direction in our future work.
>
> 4. DABertScore proposes a modification to the BertScore algorithm based on token-level difficulty weighting. Like BertScore, it is a reference-based metric. We have not found any conceptual overlap between DABertScore and our approach. Our system is a reference-free pairwise MT evaluation method that has been trained on synthetic data without using any human annotated translation quality scores. If the reviewer could clarify the conceptual overlap between our method and DABertScore we would be happy to include those comparisons in our paper.
>
>
> [1] Amplayo, R.K., Liu, P.J., Zhao, Y., & Narayan, S. (2022). [SMART: Sentences as Basic Units for Text Evaluation.](https://openreview.net/forum?id=OIe3kpwl40D) ArXiv, abs/2208.01030.

---

> > ### Author Response · Authors · 2023-11-21
> > **Continuation of Response to reviewer DxQx**
> >
> > **Response to reviewer’s questions:**
> >
> > 1. We have already discussed above how our system can be used to rank multiple MT systems. Since our system also provides a probability score for each comparison the probabilities can be used instead of simple tallying better-worse predictions.
> >
> > 2. We have shown above how to get system level scores with our systems by averaging across the row of pairwise comparison probabilities. With the system-level scores, there are no contradictions. However, there is a chance of contradiction among the better-worse judgments of systems if we do not provide this system level score and rely on each pairwise comparison probabilities to make decisions. To that end, we have checked if there is any inconsistency among the pairwise system ranking generated by our system. Among the 14 language pairs on the DA20 datasets we have only found inconsistency on 7 of them. On the following table we show the percentage of triples that show inconsistency. We note the inconsistency level is less than one percent for all the 7 language pairs.
> >
> > | zh-en | en-de | ru-en | ta-en | en-zh | pl-en | en-pl | Avg   |
> > | ----- | ----- | ----- | ----- | ----- | ----- | ----- | ----- |
> > | 0.54% | 0.14% | 0.91% | 0.27% | 0.91% | 0.41% | 0.27% | 0.49% |

---

> > ### Comment · Reviewer_DxQx · 2023-11-23
> > **Further Comments.**
> >
> > Thank you for your response. To further clarify, I have the following specific requests and suggestions:
> >
> > 1.  In your tables, you've mentioned COMET-QE, but it's not clear which variant (base, large, or XXL) is used. Could you specify this and provide the scores for the remaining variants?
> > 2.  Please include the results of T5SCORE-XLSUP in the two referenced tables for a comprehensive comparison.
> > 3.  The paper should be revised to address the main weaknesses (system ranking) identified by myself and Reviewer Qwxh.
> > 4.  I believe that COMET-QE, as a reference-free baseline, is already simple and effective. Your proposed method appears to require excessive effort for implementation without showing significant advancements. Please further discuss it.
> > 5.  Since your training and testing are based on the wmt20-da dataset, could you also provide the testing results for the wmt22-da datasets for a more current evaluation?
> > 6.  As mentioned in my first review, while it's not mandatory, including insights from recent publications could greatly enhance the robustness and relevance of your method.
> > 7.  Like your title, both DABertscore and the proposed method improve MT evaluation by **inter-system comparison**.  A detailed discussion on this aspect would be beneficial for readers to gain deeper insights into this line of research.
> > 8.  I have updated my original review to make the identified weaknesses and suggestions clearer.

---

> ### Author Response · Authors · 2023-11-23
> **Response to reviewer DxQx's further comments**
>
> Thank you for your further questions.
>
> 1. We chose the WMT20-COMET-QE-DA model for evaluation on DA20 and MQM20 datasets. We chose WMT21-COMET-QE-MQM for evaluation on MQM21 and MQM22 datasets.
> Both of these models are available from the [COMET library](https://unbabel.github.io/COMET/html/models.html). Models of different scales (ie base,large,XXL) are not available for COMET-QE.
> 2. We have updated the tables.
> 3. We will revise our paper to include discussion of system-level evaluation.
> 4. COMET-QE is supervised, It was trained on human annotations of machine translation quality scores (ie Direct Assessment and MQM scores). In contrast our models were trained on weakly supervised synthetic dataset. Our pairwise-comparison based approach enables us to create this large synthetic dataset. Thus, even though system-level evaluation with our comparison-based approach is more complicated than with score-based approach (ie COMET-QE), we believe the fact that our approach does not require costly human annotations for training makes up for it.
> 5. In the following table we show the system-level Pearson correlation of our system and the baselines on the MQM22 dataset.
>
> |                  | en-de | zh-en | en-ru | avg   |
> | ---------------- | ----- | ----- | ----- | ----- |
> | COMET-QE         | 0.48  | 0.544 | 0.468 | 0.497 |
> | COMET-Kiwi       | 0.592 | 0.795 | 0.763 | 0.717 |
> | Unite            | 0.509 | 0.791 | 0.779 | 0.693 |
> | Comparator-Base  | 0.79  | 0.74  | 0.79  | 0.773 |
> | Comparator-Large | 0.87  | 0.84  | 0.92  | 0.877 |
> | Comparator-XXL   | 0.92  | 0.85  | 0.95  | 0.907 |
>
> 6. We again note that the xCOMET paper was released after we submitted our paper to ICLR and the SLIDE paper was released only 12 days before our submission. Thus including insights from these two publications is not relevant to our current submission to ICLR.
> 7. We are still confused as we do not find any **inter-system comparison** that is similar to our approach in the DABertscore paper. Similar to Bertscore, DABertscore is a reference based approach, where machine translations are compared with the reference sentence to generate token similarity scores. Machine translations from two systems are not compared by DABertScore.
> 8. Thanks a lot for your valuable feedback.

---

### Meta-Review · Area_Chair_oKg7 · 2023-12-10

**Metareview:**

This paper proposes a novel reference-free machine translation evaluation method that directly compares the two hypotheses from two systems by pre-trained language models, e.g., mT5.

All reviewers admit the proposed method is interesting and empirical part is convincing.

Rui

**Justification For Why Not Higher Score:**

n/a

**Justification For Why Not Lower Score:**

n/a

---

### Decision · Program_Chairs · 2024-01-16

Accept (spotlight)